# Venous Thromboembolism in Patients with Inflammatory Bowel Disease: The Role of Pharmacological Therapy and Surgery

**DOI:** 10.3390/jcm9072115

**Published:** 2020-07-04

**Authors:** Alfredo Papa, Antonio Tursi, Silvio Danese, Gianludovico Rapaccini, Antonio Gasbarrini, Valerio Papa

**Affiliations:** 1Internal Medicine and Gastroenterology Department, Fondazione Policlinico Universitario A. Gemelli, IRCCS, 00168 Roma, Italy; gianludovico.rapaccini@unicatt.it (G.R.); antonio.gasbarrini@unicatt.it (A.G.); 2Università Cattolica del S. Cuore, 00168 Roma, Italy; valerio.papa@unicatt.it; 3Servizio di Gastroenterologia Territoriale ASL BAT, Barletta-Andria-Trani, 76123 Barletta, Andria, Trani, Italy; antotursi@tiscali.it; 4Humanitas Clinical and Research Center, IRCCS Rozzano, 20089 Milan, Italy; silvio.danese@hunimed.eu; 5Department of Biomedical Sciences, Humanitas University, Rozzano, 20089 Milan, Italy; 6Digestive Surgery Department, Fondazione Policlinico Universitario A. Gemelli, IRCCS, 00168 Roma, Italy

**Keywords:** inflammatory bowel disease, venous thromboembolism, infliximab, tofacitinib, surgery

## Abstract

Patients with inflammatory bowel disease (IBD) have an increased risk of venous thromboembolism (VTE). Alongside the traditional acquired and genetic risk factors for VTE, patients with IBD have pathogenic and clinical peculiarities that are responsible for the increased number of thromboembolic events occurring during their life. A relevant role in modifying this risk in a pro or antithrombotic manner is played by pharmacological therapies and surgery. The availability of several biological agents and small-molecule drugs with different mechanisms of action allows us to also tailor the treatment based on the individual prothrombotic risk to reduce the occurrence of VTE. Available review articles did not provide sufficient and updated knowledge on this topic. Therefore, we assessed the role of each single treatment, including surgery, in modifying the risk of VTE in patients with IBD to provide physicians with recommendations to minimize VTE occurrence. We found that the use of steroids, particularly if prolonged, increased VTE risk, whereas the use of infliximab seemed to reduce such risk. The data relating to the hypothesized prothrombotic risk of tofacitinib were insufficient to draw definitive conclusions. Moreover, surgery has an increased prothrombotic risk. Therefore, implementing measures to prevent VTE, not only with pharmacological prophylaxis but also by reducing patient- and surgery-specific risk factors, is necessary. Our findings confirm the importance of the knowledge of the effect of each single drug or surgery on the overall VTE risk in patients with IBD, even if further data, particularly regarding newer drugs, are needed.

## 1. Introduction

Chronic inflammatory bowel diseases (IBD), which include Crohn’s disease (CD) and ulcerative colitis (UC), primarily affect the intestine. They are burdened by numerous extraintestinal complications, with venous thromboembolism (VTE) representing one of the most feared complications owing to its consequences on morbidity and non-negligible mortality [1,2,3,4]. Recent meta-analyses reported an almost three-fold increased risk of deep vein thrombosis (DVT) and pulmonary embolism (PE) in patients with IBD compared to the general population [5,6,7]. This increased VTE risk was confirmed in hospitalized pediatric IBD patients who showed a relative risk (RR) of 2.36 (95% confidence interval (CI): 2.15–2.58) compared to healthy controls [8]. Additionally, existing data indicate a significant 2.5-fold higher odds of mortality associated with VTE-related hospitalizations than with non-VTE-related hospitalizations for IBD patients [9]. The pathogenesis of VTE in IBD patients is multifactorial, involving both inherited and acquired risk factors (see Table 1) [10,11]. However, a crucial role is played by disease activity on which many other prothrombotic factors depend. Although this review focuses on venous thromboembolic manifestations, we want to emphasize that IBD patients have an increased risk not only of VTE but also of thrombotic arterial circulation events. This is due, in particular, to the presence of active inflammation, which represents a determining factor in the genesis of atherogenic phenomena at the level of the arterial vessels. To emphasize the role of the so-called inflammatory burden, a large epidemiological study reported that IBD patients had the highest risk of VTE during a flare-up (hazard ratio (HR) of 8.4 compared with controls) [12]. Indeed, the complex interaction between inflammation and coagulation has been extensively studied in IBD patients, and it was reported that active inflammation shifts the coagulation balance towards a prothrombotic state [11]. 

To summarize the huge amount of studies on this topic, active inflammation is involved in increasing VTE risk by means of the following mechanisms: (i) increasing the plasma levels of thrombophilic factors, many of which are acute phase reactants, and decreasing the levels of natural anticoagulants; (ii) reducing fibrinolytic activity; (iii) determining several endothelial anomalies that move towards a prothrombotic state; and (iv) inducing platelet abnormalities such as thrombocytosis and increased activation and aggregation (Table 2) [11,13,14]. Thus, we can deduce that a relevant role in modifying the balance in a pro or antithrombotic manner is also played by therapies, both pharmacological treatments and surgery. Particularly, the recent availability of several biological agents and small-molecule drugs with different mechanisms of action could allow us to personalize therapy with consideration, in the decision-making process, of the risk of possible adverse events including the occurrence of VTE. However, the review articles available in the literature do not provide sufficient and updated knowledge on this topic. Therefore, the aim of this review was to assess the weight of each single treatment in modifying VTE risk in IBD patients to provide gastroenterologists with a practical guide on the use of the different drugs available and according to the thrombotic risk.

## 2. Pharmacological Therapies

As previously reported, control of disease activity could be expected to reduce VTE risk by decreasing procoagulant factors closely associated with inflammatory mediators. Additionally, several studies report that some IBD therapeutics exhibit intrinsic anticoagulant properties. Contrarily, some drugs have been associated with an increased VTE risk. In the following sections, we analyze the VTE risk of each of the pharmacological treatments used for IBD management, focusing on the most common and recent treatments such as corticosteroids (CSs), biologics, and tofacitinib (TOF).

### 2.1. Mesalazine

Mesalazine, a mainstay of IBD therapy, induces a wide array of modulatory activities, including inhibition of platelet activation [15]. In fact, platelets isolated from the circulation of IBD patients receiving mesalazine or olsalazine displayed reduced spontaneous and thrombin-induced platelet activation in vitro and low expression of P-selectin in vivo [15]. A plausible explanation is that reduced activation resulted in less interaction with endothelial and inflammatory cells and, subsequently, diminished release of proinflammatory mediators, leading to downregulation of gut inflammation. Overall, the available evidence shows that mesalamine could reduce the VTE risk, although specifically designed studies are not available.

### 2.2. Azathioprine and 6-Mercaptopurine (6-MP)

Azathioprine and its active metabolite 6-mercaptopurine are the two of the most widely used immunosuppressant agents in IBD. Besides their well-known effects on the immune system, these agents also inhibit adenosine diphosphate-induced platelet aggregation in vitro [16]. and this blockade inhibits formation of platelet-leukocyte aggregates, which play a role in the pathogenesis of IBD and thrombosis [17]. Thus, their use could reduce the VTE risk even if there are no data confirming this effect.

### 2.3. Corticosteroids (CSs)

CSs induce a procoagulant state as reported by a recent randomized, placebo-controlled study including healthy individuals who received either oral prednisolone (0.5 mg/Kg/day) or a placebo for 10 days [18]. Prednisolone increased peak thrombin, velocity index, plasminogen-activator inhibitor type-1 (PAI-1) and von Willebrand factor (VWF) compared with the placebo group, suggesting that CS treatment may contribute to thromboembolic risk [18]. In 2013, a Danish nationwide population-based case-control study assessed the association between CSs and VTE. Of 38,765 VTE cases included in the study, 4119 (10.6%) had autoimmune diseases including IBD [19]. The authors found an adjusted incidence rate ratio (IRR) of 2.31 for VTE in patients using CSs with a temporal gradient of increased risk in new use that declined with time after multiple adjustments for disease severity [19]. The temporality of the association, whereby the strongest effect occurs at the beginning of therapy and the absence of effect after discontinuation, was in line with the effect of the risk factor considered in this study (i.e., CSs) on coagulation. Moreover, taking intestinal acting CSs, used exclusively for the treatment of IBD patients, increased the VTE risk both among new users (adjusted IRR, 2.17; 95% CI: 1.27–3.71) and continued users (1.76; 1.22–2.56) [19]. This finding is of clinical relevance, since it demonstrates that, despite their reduced bioavailability, the systemic absorption of intestinal acting CSs is clinically significant, because it can increase VTE risk. 

Additional data showing a prothrombotic effect of steroidal treatment comes from a retrospective cohort study involving 30,456 IBD patients, 32% of whom required at least one course of CSs during the study period [20]. The rate of VTE per 1000-person years was higher among CS users than among non-CS users with a trend towards statistical significance (9% versus 4.9%, *p* = 0.080) [20]. However, after adjusting for age, male sex and Charlson comorbidity index, the authors found that compared to the year after the study period index IBD diagnosis, CS users had 4.68 greater probability of experiencing VTE (IRR = 4.68, 95% CI: 3.52–6.24, *p* < 0.001) [20]. Data from the National Surgical Quality Improvement Program were collected on 75,771 surgical patients who underwent nine common general, vascular and orthopedic operations (including colectomy) to assess the risk factors for symptomatic postoperative VTE [21]. Preoperative CS use was associated with an odds ratio of 1.87 (95% CI: 1.37–2.53) for VTE [21]. Finally, indirect evidence confirming the role of CSs in increasing VTE risk comes from a recent meta-analysis, which compared VTE risk between IBD patients receiving systemic CSs and those receiving TNF-α antagonists [22]. CS users showed a 5-fold increased VTE risk compared to anti-TNF-α agent users (OR: 0.267; 95% CI: 0.106–0.674, *p* = 0.005) [22]. Overall, these data seem to indicate that CS treatment, although it reduces the inflammatory activity, determines a significant increase in thrombotic risk, especially when compared to infliximab (IFX) therapy. Thus, physicians should always keep in mind that VTE is a possible side-effect of CSs and, consequently, all the available measures for VTE prevention should be implemented according to current recommendations [23].

### 2.4. Methotrexate (MTX)

MTX causes increased levels of homocysteine by antagonizing folic acid. Elevated fasting serum homocysteine levels are associated with an increased VTE risk. IBD patients were reported to have a higher prevalence of hyperhomocysteinemia than healthy subjects [24]. Folate deficiency has been found as an independent risk factor in the development of hyperhomocysteinemia due to malnutrition, malabsorption and/or folate antagonist therapy [24]. Therefore, IBD patients prescribed MTX should receive folate supplementation with the aim of maintaining normal homocysteine levels to reduce the VTE risk.

### 2.5. Thalidomide

Thalidomide is an immunomodulatory drug approved for the treatment of multiple myeloma (MM) and for other inflammatory conditions including CD [25]. It causes a significant increase in CD62P expression on platelets, thrombin-antithrombin complexes (TAT) levels, factor VIII activity and soluble trombomodulin (sTM) concentration in patients with MM [26]. Therefore, these findings demonstrate that thalidomide is able to increase platelet activation and to enhance the hypercoagulable state in MM patients. Thalidomide is associated with an increased prevalence of venous thrombotic complications, also in nonmalignant disease. A meta-analysis concluded that the RR for thromboembolism was increased with thalidomide alone (RR 2.6), steroids alone (RR 2.8) and in combination (RR 8.33) [27]. Thus, CD patients receiving thalidomide, especially with concurrent steroids, should be followed closely to reduce their VTE risk and should be considered for VTE prophylaxis.

### 2.6. Infliximab (and Other Anti-Tumor Necrosis Factor (TNF)-α Agents)

Data from the British Society for Rheumatology Biologics Register, a national prospective observational cohort study of biological safety in patients with rheumatoid arthritis (RA), compared the incidence of VTE between 11,881 anti-TNF-α- (IFX, adalimumab, and etanercept) and 3673 nonbiological disease-modifying antirheumatic drug (nbDMARD)-treated patients [28]. Overall, there was no difference in VTE rates between the two treatment groups (adjusted HR, 0.8; 95% CI: 0.5–1.5). Additionally, the risk was similar across all anti-TNF-α agents [28]. Moreover, a retrospective study including 15,100 IBD patients was conducted to evaluate the effects of biologic, CS and combination therapies (biologics plus CS) on VTE risk during a 12-month follow-up period [29]. The authors reported that the treatment of active IBD with biologic agents reduced the VTE risk (OR 0.21, 95% CI: 0.05–0.87) compared with treatment with CS alone. Moreover, the combination therapy was associated with the same VTE risk as that of CS alone. Thus, we considered the latter as responsible for the observed increased VTE risk [29]. 

Further support to the hypothesis of an anti-thrombotic effect of anti-TNF-α agents comes from the abovementioned meta-analysis comparing VTE risk in IBD patients as a complication of systemic CS or anti-TNF-α therapies [21]. Overall, anti-TNF-α treatment resulted in a 5-fold decreased VTE risk compared to steroid medications [21]. A further large nationwide observational study conducted in the United States assessed the VTE risk of IBD patients treated with TNF-α inhibitors or nonbiologic immunomodulating agents such as azathioprine, 6-MP, MTX and cyclosporine [30]. Although, overall, there was no statistically significant difference in the VTE risk between the two groups, a subgroup analysis showed that treatment with TNF-α inhibitors was associated with VTE risk reduction in CD patients aged <45 years (HR 0.62 (95% CI: 0.44–0.86) and 0.55 (95% CI: 0.34–0.87)), respectively [30]. Several studies report that the antithrombotic effect of TNF-α antagonists is not only due to the reduction of inflammation but also to other specific anticoagulant actions. Indeed, IFX normalizes hemostatic parameters and reduces the levels of circulating microparticles and prothrombotic sCD40L in CD patients [31,32]. Recently, Bollen et al. investigated the effect of IFX induction therapy on the hemostatic profile and clot lysis profile of 103 IBD patients starting IFX therapy and 113 healthy controls [33]. On the IFX induction treatment, the clot lysis profile normalized in the responders to therapy, suggesting that IFX treatment can normalize activated hemostatic profiles of IBD patients [33].

### 2.7. Vedolizumab (VDZ)

VDZ is a monoclonal antibody that inhibits the interaction between α4β7 integrin and mucosal addressing cell adhesion molecule-1 (MAdCAM-1), which is selectively expressed by the vascular endothelium in the gastro-intestinal tract. This drug has been approved for moderate-to-severe UC and CD. Data on the risk of cardiovascular events (CVEs) and VTE in patients treated with VDZ are scarce. Pivotal studies in UC (GEMINI-1) and CD (GEMINI 2) found 1/895 (0.11%) death due to acute coronary syndrome and 1/1115 (0.09%) death due to myocarditis in a male patient with a history of DVT [34,35]. Neither the post hoc analysis of GEMINI 2 and GEMINI 3 trials, nor long-term trials of GEMINI LTS, found cases of CVEs or VTE even when VDZ was associated with CSs [36,37,38]. A recent VARSITY trial, which compared VDZ with adalimumab, recorded only one case out of 383 (0.26%) patients treated with VDZ with superficial thrombophlebitis [39]. Finally, a recent real-life analysis conducted in the United States reported 10 pulmonary embolism events (1% of the VDZ population) [40]. In conclusion, VDZ has a low overall VTE risk.

### 2.8. Ustekinumab (UST)

UST is a monoclonal antibody blocking the p40 subunit of interleukin-12 and interleukin-23 and has been approved for use in the treatment of psoriasis, psoriatic arthritis and CD. Although no specific trials have been designed to assess the association between UST and VTE occurrence, in pivotal trials with 1369 CD patients neither CVEs nor VTE events were recorded [41]. In the first extension of these studies, 718 patients were followed up for 2 years and two deaths due to CVEs were observed [42]. In a further 3-year follow-up extension, 567 patients were included and only one CVE was reported [43]. Overall, the incidence of CVEs and VTE was 0.22% at the 3-year follow-up [43]. Recently, this finding has been confirmed by a retrospective observational Belgian study, which recorded only one case of DVT in a population of 152 CD patients (0.65%) treated with UST [44].

### 2.9. Tofacitinib (TOF)

TOF, an orally administered small-molecule Janus Kinase (JAK)-1 and -3 inhibitor, is approved for the treatment of UC, rheumatoid arthritis, psoriatic arthritis and chronic plaque psoriasis. A meta-analysis of randomized controlled trials (RCTs) assessed the association of TOF usage with CVEs including both major CVEs and VTE, and all-cause mortality in adult patients with immune-mediated inflammatory diseases (IMIDs) [45]. Overall, 27 RCTs (randomizing 13,611 patients) comparing TOF with placebo or simultaneously two dose regimens of TOF (5 mg versus 10 mg b.d.) were included in this meta-analysis [45]. Compared with placebo, there was no increased risk of CVEs and all-cause mortality in IMID patients receiving TOF in the short term, whereas 10 mg b.d. TOF appeared to be associated with reduction in CVE and all-cause mortality risks, except VTEs, relative to twice-daily dosing of 5 mg TOF. Indeed, an increasing trend in VTE risk (OR = 1.47, 95% CI: 0.25–8.50) was identified compared with the 5 mg regimen [45]. Additionally, a Food and Drug Administration postmarketing requirement safety study designed to evaluate the long-term risk of major adverse cardiovascular events and malignancy in patients with rheumatological diseases found that the frequency of PE in the 10 mg b.d. TOF arm was higher than that of the TNF-α-inhibitor comparator arm [46]. 

Moving to IBD patients, Sandborn et al. reported post hoc analysis data on the incidence of VTE in 1157 UC patients treated with TOF [46]. Five VTE cases were reported (4 PE and 1 DVT) in patients treated with 10 mg b.d. TOF, whereas 2 PE cases and 2 DVT cases were observed in the placebo-treated patients [47]. The incidence rate (patients with events/100 patient-year; 95% CI) was 0.04 (0.00–0.23) for DVT and 0.16 (0.04–0.41) for PE [47]. Recently, Kotze et al. described two additional cases: a 68-years old male with extensive UC who started TOF 10 mg b.d. and after 51 days of treatment was diagnosed with acute bilateral DVT completely resolved with enoxaparin treatment, and a 47-year-old female with extensive UC who had a fatal acute myocardial infarction after 2 weeks from starting TOF 10 mg b.d. [48]. Neither patient had risk factors for thrombosis or previous history for cardiovascular or VTE. Obviously, these analyses were limited by the small sample size and short drug exposure. Therefore, further studies assessing the occurrence of VTE in patients treated with TOF are needed before drawing definitive conclusions. Until then, given that VTE has been determined as a potential risk of TOF therapy, gastroenterologists should individualize treatment by considering all risk factors for VTE for each patient. Until now, despite the absence of data that directly compare treatment with TOF (or other anti-JAK) to biological therapies (i.e., IFX) in terms of the incidence of VTE, it is advisable to use TOF in patients without known risk factors for VTE (see Table 1).

### 2.10. Upadacitinib

Recently, upadacitinib, an oral selective inhibitor of JAK-1, was used in a phase 2b study as induction therapy for 250 adults with moderately to severely active UC [49]. Among patients taking 45 mg upadacitinib once daily, one patient developed PE and DVT (diagnosed at 26 days after treatment discontinuation).

### 2.11. Filgotinib

Filgotinib is also a selective JAK-1 inhibitor. It is under investigation for various indications such as rheumatoid arthritis, psoriatic arthritis, ankylosing spondylitis and CD [50]. There is no evidence of an increased risk of VTE in patients being treated with filgotinib or significant differences compared to other molecules of the same class [51].

## 3. Surgery

Abdominal surgery is a frequent intervention for IBD patients. In fact, during the natural course of IBD, most CD patients undergo at least one surgical intervention (both intestinal resection or for perianal disease) and approximately 10% of UC patients undergo colectomy, usually with a subsequent construction of an ileo-anal pouch. However, surgery in IBD patients carries a significant VTE risk, despite adequate prophylaxis implementation [52,53]. In fact, a systematic review including 38 studies reported that the postoperative rates of VTE in IBD patients ranged from 0.6% to 8.9% [52]. The most important patient-specific risk factors for postoperative VTE are UC, increasing age and obesity, whereas surgery-specific risk factors include open surgery, emergent surgery and ileostomy creation [52]. Furthermore, a recent retrospective study involving 434 UC patients assessed the risk of portomesenteric venous thrombosis (PMVT) following colectomy [54]. Although postoperative VTE-prophylaxis was administered to 98.5% of inpatients, PMVT still developed in 36 (8.3%) patients [54]. The majority of PMVT cases occurred after subtotal colectomy, and preoperative C-reactive protein values > 45 mg/L were significantly associated with the onset of PMVT (*p* = 0.01) [54]. 

Regarding the prevention of postoperative VTE, a retrospective review of patient data obtained from the American College of Surgeons National Surgical Quality Improvement Program reported that the following actions can reduce the incidence of VTE in IBD patients: correcting preoperative coagulopathy and/or anemia, improving nutritional status, reducing steroid use, operating early to avoid emergency surgery and limiting anesthesia time [55]. Additionally, a retrospective review including 75,620 patients who underwent colorectal resection for diverticulitis, cancer and IBD reported 30-day rates of PE or DVT of 2.4%, 2.9% and 3.1%, respectively (*p* < 0.001) [55]. These data indicated that IBD patients (particularly UC patients) had a significantly increased risk for postoperative VTE compared to patients with colorectal cancer (CRC) and, consequently, the postdischarge VTE prophylaxis recommendations for IBD patients should mirror those for patients who had undergone operations for CRC. Thus, in clinical practice, out-of-hospital prophylaxis for IBD patients undergoing surgery should be extended for at least 4 weeks [56,57]. Further attention should be paid to patients with IBD undergoing CRC surgery. In fact, by comparing the hospital outcomes of CRC surgery between IBD and non-IBD patients, using data extracted from the National Inpatient Sample (2008–2012) and Nationwide Readmissions Database (2013), we found that IBD patients were more likely to develop DVT (adjusted odds ratio: 2.42, 95% CI: 1.36, 4.28) when compared to non-IBD patients with CRC [58]. In conclusion, maximum effort must be made by gastroenterologists and surgeons, especially in the phase prior to surgery, to minimize the risk factors for VTE.

## 4. Conclusions

Patients with IBD have an increased VTE risk, particularly in the active disease phase. Therefore, all physicians involved in the management of IBD patients should implement VTE prevention, which includes personalization of IBD treatment based on the individual’s thrombotic risk and on knowledge of the effect of each single drug, or surgery, in modifying the overall risk (Table 3). Therefore, the use of CSs, particularly if prolonged, increases VTE risk, whereas the use of some biological therapies, including IFX, seems to reduce such risk. The data relating to the hypothesized prothrombotic risk of TOF are insufficient to draw definitive conclusions. Moreover, surgery has an increased prothrombotic risk. Therefore, implementing measures to prevent VTE, not only with pharmacological prophylaxis, but also by reducing patient- and surgery-specific risk factors, is necessary.

## Figures and Tables

**Table 1 jcm-09-02115-t001:** Acquired risk factors for VTE in IBD patients and modalities for their prevention and/or treatment.

Risk Factor	Prevention/Treatment Modality
**Hyperhomocysteinemia**	Correction of vitamin deficiency (vitamin B12, vitamin B6, and folic acid)
**Dehydration**	Provide adequate hydration
**Prolonged immobilization**	Early mobilization, especially after surgery; graduated compression stockings or pneumatic devices
**Infections**	Timely diagnosis and treatment of infections
**Indwelling catheters**	Limit the use of venous catheters; when possible, administer oral and enteral nutrition
**Obesity**	Encourage weight loss (diet and exercise)
**Hospitalization**	VTE prophylaxis also for IBD patients admitted for non-IBD related reasons
**Smoking**	Programs for smoking cessation
**Oral contraceptive use**	Advise alternative methods of contraception
**Previous VTE**	Search for genetic and acquired risk factors for VTE and administer prophylaxis if necessary

Abbreviations: venous thromboembolism, VTE; inflammatory bowel disease, IBD.

**Table 2 jcm-09-02115-t002:** Abnormalities of hemostatic parameters observed in patients with inflammatory bowel disease.

**Abnormalities of Coagulation**
↑ Fibrinogen
↑ Factors V, VIII, IX
↑ Prothrombin fragment 1 + 2, fibrinopeptide A and B, TAT complex
↓ Factor XIII/subunit A factor XIII
↓ Protein C, protein S, antithrombin
↓ TFPI
**Abnormalities of Platelets**
↑ Number, activation, aggregation
**Abnormalities of Fibrinolysis**
↓ tPA
↑ PAI, TAFI
↑ D-dimer, FDP, FgDP
**Endothelial Abnormalities**
↑ Circulating thrombomodulin, ECPR, and von Willebrand factor
↓ Tissue thrombomodulin and EPCR
**Nutritional Abnormalities**
↑ Homocysteinemia, lipoprotein A
↓ Vitamin B6, Vitamin B12, folates
**Immunological Abnormalities**
Antibodies: antiphospholipid, antiprotein S, antiendothelial cells, anti-tPA

Abbreviations: inflammatory bowel disease, IBD; thrombin anti-thrombin, TAT; tissue factor pathway inhibitor, TFPI; tissue-type plasminogen activator, tPA; plasminogen activator inhibitor PAI; thrombin-activatable fibrinolyis inhibitor, TAFI; fibrin degradation products, FDP; fibrinogen degradation products, FgDP; endothelial protein C receptor, EPCR.

**Table 3 jcm-09-02115-t003:** Therapies for patients with inflammatory bowel disease and risk of venous thromboembolism.

Drug	Route of Administration	Therapeutic Indication	Risk of VTE	Comments
**Mesalazine**	Oral and topical	Mild-to moderate UC Prevention of postoperative recurrence of CD	Reduced	Specially designed studies are not available
**Azathioprine/6-mercaptopurine**	Oral	UC and CD maintenance of remission	Reduced	Specially designed studies are not available
**CSs**	Oral, intramuscular, intravenous, topical	Moderately to severely active UC or CD	Increased	Increased VTE risk also for intestinal acting steroids
**MTX**	Subcutaneous or intramuscular	CD maintenance of remission	Unclear	MTX needs folate supplementation to reduce hyperhomocysteinemia
**Thalidomide**	Oral	Steroid-resistant or steroid-dependent CD	Increased	VTE risk significantly increased when associated with CSs; VTE prophylaxis should be considered
**Infliximab (and other anti-TNF-α)**	Intravenous (IFX)Subcutaneous (ADA and GOL)	Steroid-resistant or steroid-dependent UC and CD	Reduced	Increased when IFX is associated with CS
**Vedolizumab**	Intravenous	Steroid-resistant or steroid-dependent UC and CD	Reduced	Paucity of data. Association with CS does not increase the risk
**Ustekinumab**	Intravenous (induction) followed by subcutaneous (maintenance)	Steroid-resistant or steroid-dependent CD	Reduced	Paucity of data
**Tofacitinib**	Oral	Moderately to severely active UC refractory to standard treatments	Unclear	Data from RCTs and observational studies are not sufficient to provide conclusive advice

Abbreviations: corticosteroids, CS; Crohn’s disease, CD; ulcerative colitis, UC; infliximab, IFX; adalimumab, ADA; golimumab, GOL; methotrexate, MTX; venous thrombo-embolism, VTE; tumor necrosis factor, TNF; randomized controlled trial, RCT.

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
