# Peer review of "Venous Thromboembolism in Patients with Inflammatory Bowel Disease: The Role of Pharmacological Therapy and Surgery"

_jcm, 2020, doi:10.3390/jcm9072115_

Round 1
Reviewer 1 Report
- In section 2.3 Corticosteroids, it would be valuable to provide the information that prednisolone increased peak thrombin, velocity index, PAI‐1 and VWF as compared with the placebo group in healthy volunteers (Majoor CJ et al. J Thromb Haemost 2016;14:716—23)
- Section 2.5 Thalidomide, please add that the drug causes a significant increase in CD62P expression on platelets, TAT levels, factor VIII activity and sTM concentration in patients with multiple myeloma (Robak M et al. Med Oncol 2012;29:3574—80).
- The conclusion (Section 4) that “biological therapies … seem to reduce such risk” should be modified, since there are apparently signals that tofacitinib and perhaps upadacitinib do not. One could write “some biological therapies, including IFX, seem to reduce such risk”
Minor comments
- Abstract: Remove the periods after the citations that are in the middle of the sentence “Although the pathogenesis of VTE in IBD patients is multifactorial, involving both inherited [10]. and acquired risk factors [11]. (see Table 1), …”
- Table 2. It should be Prothrombin fragment 1 + 2 and it should be antithrombin (without the “III”, which was deleted about 3 decades ago, since there is since then only one antithrombin). What is ECPR?
- Page 4, spell out TNF the first time.
- Page 5,Infliximab section, change anti-TNF-α inhibitors to TNF-α inhibitors
- Vedolixumab – consider changing “thromboembolic pulmonary events” to “pulmonary embolism events” if that is what is meant.
- Upadacitinib – delete “was reported”
Author Response
Point-by-point response to the reviewer’s comments.
Authors are grateful for the valuable comments of the Reviewers. We tried our best to address each comment in the following report. Possible corrections based on the comments were done and a revised manuscript was attached. We hope that our replies and modifications will meet the Reviewers and Editor’s requirements.
Reviewer 1
Comments and Suggestions for Authors
- In section 2.3 Corticosteroids, it would be valuable to provide the information that prednisolone increased peak thrombin, velocity index, PAI‐1 and VWF as compared with the placebo group in healthy volunteers (Majoor CJ et al. J Thromb Haemost 2016;14:716—23)
We added in the section 2.3 (page 3-4, lines 98-102) some findings about the procoagulant effects of prednisolone as reported in the paper by Majoor CJ et al. (J Thromb Haemost 2016;14:716—23) that we added among the references.
- Section 2.5 Thalidomide, please add that the drug causes a significant increase in CD62P expression on platelets, TAT levels, factor VIII activity and sTM concentration in patients with multiple myeloma (Robak M et al. Med Oncol 2012;29:3574—80).
We added in the section 2.5 (page 4, lines 143-147) some findings about thalidomide’s effects on platelets’ and coagulation’s parameters as reported in the paper by Robak M et al. that we added among the references.
- The conclusion (Section 4) that “biological therapies … seem to reduce such risk” should be modified, since there are apparently signals that tofacitinib and perhaps upadacitinib do not. One could write “some biological therapies, including IFX, seem to reduce such risk”
We modified the conclusion (page 7, line 285-286) as suggested by the reviewer.
Minor comments
- Abstract: Remove the periods after the citations that are in the middle of the sentence “Although the pathogenesis of VTE in IBD patients is multifactorial, involving both inherited [10]. and acquired risk factors [11]. (see Table 1), …”
Introduction (page 1, line 38). We modified the sentence by inserting the references at the end.
- Table 2. It should be Prothrombin fragment 1 + 2 and it should be antithrombin (without the “III”, which was deleted about 3 decades ago, since there is since then only one antithrombin). What is ECPR?
We modified Table 2 according to the reviewer’s suggestion; in particular, we reported Prothrombin fragment 1+2 and modified antithrombin III in antithrombin. The meaning of ECPR was specified in the abbreviations after the table 2.
- Page 4, spell out TNF the first time.
Page 5, line 153. We spell out TNF for the first time.
- Page 5, Infliximab section, change anti-TNF-α inhibitors to TNF-α inhibitors
Page 5, line 173. We changed anti-TNF-α inhibitors to TNF-α inhibitors
- Vedolizumab – consider changing “thromboembolic pulmonary events” to “pulmonary embolism events” if that is what is meant.
Page 5, line 194. We changed “thromboembolic pulmonary events” to “pulmonary embolism events”.
- Upadacitinib – delete “was reported”
Page 6, line 242. We deleted “was reported” at the end of the sentence.
Reviewer 2 Report
This review by Papa et al. entitled “Venous Thromboembolism in Patients with Inflammatory Bowel Disease: The Role of Pharmacological Therapy and Surgery” focused on the association of active inflammation with venous thromboembolism (VTE) in patients with inflammatory bowel disease (IBD). iNDEED, Inflammation shifts the coagulation balance towards a prothrombotic state, thereby favouring venous thromboembolic complications. Authors also emphasized the increased risk of VTE associated with therapy OF ibd, particularly with steroids, methotrexate and thalidomide.
Vascular risk in IBD patients also encompasses the involvement of arterial tree, where inflammation is a prominent player. Authors are kindly requested to bring this topic to the fore in the Discussion, to emphasize the complex systemic involvement of both venous and arterial trees in IBD patients. This further strengthens the importance of vascular risk in IBD patients.
Author Response
Point-by-point response to the reviewer’s comments.
Authors are grateful for the valuable comments of the Reviewer. We tried our best to address each comment in the following report. Possible corrections based on the comments were done and a revised manuscript was attached. We hope that our replies and modifications will meet the Reviewer's requirements.
Reviewer 2
This review by Papa et al. entitled “Venous Thromboembolism in Patients with Inflammatory Bowel Disease: The Role of Pharmacological Therapy and Surgery” focused on the association of active inflammation with venous thromboembolism (VTE) in patients with inflammatory bowel disease (IBD). Indeed, inflammation shifts the coagulation balance towards a prothrombotic state, thereby favouring venous thromboembolic complications. Authors also emphasized the increased risk of VTE associated with therapy of IBD, particularly with steroids, methotrexate and thalidomide.
Vascular risk in IBD patients also encompasses the involvement of arterial tree, where inflammation is a prominent player. Authors are kindly requested to bring this topic to the fore in the Discussion, to emphasize the complex systemic involvement of both venous and arterial trees in IBD patients. This further strengthens the importance of vascular risk in IBD patients.
We agree with the Reviewer that IBD patients have an increased risk of arterial thrombotic events since active inflammation determines a procoagulant state. Although this review focuses on venous thromboembolic manifestations, we added in the introduction (page 1, lines 39-42) that IBD patients have an increased risk not only of venous thromboembolic events but also of arterial circulation thrombotic events. This is particularly due to the presence of active inflammation which represents a determining factor in the genesis of atherogenic phenomena at the level of the arterial vessels.